# Detrimental Effects of Disempowering Climates on Teaching Intention in (Physical Education) Initial Teacher Education

**DOI:** 10.3390/ijerph20010878

**Published:** 2023-01-03

**Authors:** Ginés David López-García, Antonio Granero-Gallegos, María Carrasco-Poyatos, Rafael Burgueño

**Affiliations:** 1Department of Education, University of Almeria, 04120 Almeria, Spain; 2Health Research Centre, University of Almeria, 04120 Almeria, Spain

**Keywords:** teacher education, choose teaching as a profession, basic psychological needs, controlled forms of motivation, dark side, self-determination theory

## Abstract

Previous research has looked at the positive consequences generated by teacher-generated climates on the motivational experiences of pre-service teachers. However, there is scant research focusing on the adverse motivational consequences that affect the perceptions of future teachers during the training process. The objective of this study was to explore the dark side of Duda’s multidimensional conceptualization, its influence on academic engagement, and the intention of pre-service teachers to be educators. A total of 1,410 university students in initial teacher training (including physical education pre-service teachers) (59.6% women; 40.3% men; 0.1% other; *M*_age_ = 23.85; *SD* = 5.13) participated. The following scales were used: disempowering motivational climate, frustration of basic psychological needs, academic motivation, academic engagement, and the intention to choose teaching. The results of the structural equation model with latent variables show the positive prediction of the disempowering climate on the dark side and its negative influence on the intention to be a teacher. Controlled motivation preceded by academic engagement significantly mediates the relationship between a disempowering climate and the intention to be a teacher, increasing the total effect on the latter variable. Therefore, this research highlights for both teachers and researchers the impact of a disempowering motivational style, as well as its influence on the dark side as a negative promoter in trainee teachers regarding their intention to become teachers.

## 1. Introduction

The high rate of teacher dropout during the first years of professional work has heightened the level of concern within the initial teacher training programs [1]; these are undertaken at a complicated and sensitive time and have a significant impact on the development of future teachers [2,3,4]. Among the reasons for the problems that arise are the abandonment of the profession during the first few years [5], and the decision to opt for other careers with better job opportunities [6]. Given this growing concern, authors such as Hong et al. [1] have identified the main causes of teaching abandonment, one of which is a lack of initial motivation to pursue the profession [7]. Accordingly, the prior intention towards teaching behaviour (i.e., the intention to choose teaching) has been considered in recent literature [8] as one of the relevant cognitive consequences for completing a teacher training program, as well as being necessary for practising the profession in the future. Understanding the influential factors involved in choosing teaching as a professional career is necessary to improve the quality of teaching [9]. In this regard, authors such as Fokkens-Bruinsma and Canrinus [10] mainly highlight motivation and the social-psychological environment (i.e., the educational environment) as the key outcomes influencing the choice of teaching as a career.

Recent advances in the educational field have tried to examine how socio-contextual classroom environments (i.e., the social-psychological environment) affect motivational processes, as well as the role the teacher trainer might adopt in the initial teacher training context [11,12,13,14]. In this sense, following self-determination theory (SDT) [15], various interpersonal teaching styles are recognised (e.g., autonomy support and controlling style) in the teaching intervention. Likewise, achievement goal theory (AGT) [16] recognizes the role of the teacher as among those environmental or climate factors that can influence different variables at the academic level (e.g., task-involving and ego-involving climates). In this regard, Duda and Appleton [17] suggested a conceptualization of the motivational climate integrating the dimensions of the classroom social environment of the AGT [16] and the motivational style of the SDT [15]. This new conceptualization combines AGT and SDT in a multidimensional motivational climate of greater empowering or disempowering created by the teacher, which can influence the way students think, feel and act [17,18]. Although the existence of both climates has been taken together since their conceptualization (see [17]), the disempowering motivational climate has hardly received any attention in the scientific literature [19]. The few studies that do exist [18,20] have focused on evaluating the psychometric properties of the scale, or they have not looked in detail at the development and discussion of the motivational aspects present in a disempowering motivational climate, even though they evaluated both climates [21].To date, there is no record of studies focused on the effects of a disempowering motivational climate in terms of its maladaptive and cognitive motivational consequences for students. Therefore, the present research aims to analyse the potential role that social and environmental classroom factors (i.e., a disempowering climate) might play on the intention to be a teacher, examining the possible mediating role of maladaptive motivational (i.e., the dark side of motivation) and cognitive (i.e., academic engagement) processes in the context of initial teacher training. 

### 1.1. Disempowering Motivational Climate

A disempowering climate is characterized as having a greater ego-involving and controlling style (see [17]). The controlling style [15] establishes ways of thinking, acting and behaving that are imposed by the teacher, independent of the students’ interests, while the ego-involving climate [16] focuses on criticising students’ mistakes and rewarding the most competent. On the one hand, Duda’s disempowering climate framework has been negatively related to cognitive variables (e.g., academic engagement or the intention to teach). Specifically, authors such as Guo et al. [22] have only examined the relationship between an empowering climate and academic engagement, leaving aside the possible maladaptive influence of a disempowering climate. The possible negative influence of a disempowering climate on the intention of future behaviours (i.e., the intention to choose teaching as a career) has not yet been examined, despite its influence on the teacher training context. On the other hand, a disempowering climate has been positively related to the maladaptive motivational outcomes encompassed within the SDT [17]. For example, in a study on teachers in initial training, a disempowering climate positively predicted basic psychological needs frustration [21]. This relationship between maladaptive social environments created by the teacher (i.e., perceived controlled behaviours) and negative self-determined motivation variables (e.g., basic psychological needs frustration) has been conceptualized in previous works as the dark side of motivational processes [23]. However, despite its importance in the educational field, recent studies based on the dark side [24,25] have not taken into account the possible role of the ego-involving climate, as conceptualized by Duda and Appleton [17] with regard to educational and motivational outcomes. 

### 1.2. Dark Motivational Pathway

SDT [26,27] suggests that the socio-contextual environment influences the motivational behaviours of individuals, including in the educational setting. Specifically, the perception of external pressures and the use of ego-centred coercive means by the teacher will lead to the frustration of the students’ basic psychological needs (FBPN), understood as the so-called dark side. Basic psychological needs (BPN) are viewed as the essential and universal nutrients for optimal development, growth and well-being [17,28]. In this sense, FBPN is explained as: the frustration of autonomy (i.e., the degree of pressure and internal feeling carried out by an individual), the frustration of competence (i.e., the degree of inferiority or failure to execute a task at a certain level), the frustration of relatedness (i.e., the degree of perceived loneliness and alienation) and the frustration of novelty (i.e., the degree of perceived monotony and invariability) [28,29]. Likewise, authors such as Viksi and Tilga [25] have evaluated the dark side via the trans-contextual model of motivation, comprising the least self-determined types of motivation: controlled motivation (CM) and amotivation [26]. CM is conceived as both introjected and external regulation, while amotivation is understood as the absence of self-determination and regulation towards the desired behaviour. In this regard, SDT postulates that the dark side of motivation (i.e., FBPN, CM, and amotivation) will reduce the affective, cognitive and behavioural consequences in an adaptive way within the teacher training context. In the educational context, several authors have noted the negative relationship between the dark side and cognitive outcomes such as academic engagement or the intention to teach. Specifically, academic engagement has been negatively related to the dark side: CM [30], amotivation [31] and FBPN [32]. In addition, authors such as Burgueño et al. [8] have negatively linked FPBN to the intention to teach in pre-service teachers. Although recent research has analysed the dark side in an educational context [24,25], to date, no analysis has taken into account the less self-determined end of the trans-contextual model of motivation together with FPBN in trainee teachers.

### 1.3. Academic Engagement

Academic engagement has been conceptualized as the positive affective and mental state related to academic work, involving the intention, interest and effort invested by students in the learning process [33]. Previous research [33] has operationalized academic engagement across different dimensions: vigour, dedication, and absorption. Specifically, research in the teacher training context has shown the importance of academic engagement both as a consequence of socio-contextual classroom environments [14] and as a mediator between educational outcomes [13]. Although the use of academic engagement as an antecedent of future behaviours has been evidenced [34], few studies have evaluated its possible positive influence on behavioural intention (i.e., the intention to be a teacher) [35] despite its relevance in the teacher training context [8,10,36].

### 1.4. Theory of Planned Behaviour

The intention to be a teacher is a construct created from the behavioural intention encompassed in the theory of planned behaviour [37]. Behavioural intent (i.e., the intention to choose teaching as a career) represents an immediate antecedent of a person’s degree of effort to act. Specifically, in the context of initial training, the intention to be a teacher reflects the degree of planning and effort that pre-service teachers employ in working as teachers. Likewise, according to Ajzen [37,38], the intention to choose teaching as a career is influenced by: (i) the attitude towards the behaviour, that is, a positive or negative behavioural assessment of teaching; (ii) the subjective norm, i.e., the individual’s beliefs, which indicates whether there is social pressure to engage in certain behaviours; (iii) the perceived behavioural control, i.e., the degree of perceived ease or complexity associated with performing the future behaviour. Thus, previous intentions towards teaching behaviour are considered relevant cognitive consequences for the future professional practice of trainee teachers.

### 1.5. The Present Study

Given the importance of the intention to teach in pre-service teachers, both during their training and while practising professionally as teachers, a predictive analysis is required of the motivational and engagement variables affecting the intention to teach. Furthermore, to the best of our knowledge, no analysis of the mediating role played by the dark side of motivation between a disempowering motivating climate and the intention to teach in pre-service teachers has been addressed in the scientific literature. Taking into account the postulates of the SDT and AGT, as well as a review of previous studies, a hypothesized model (see Figure 1) was created to examine the above-mentioned relationships. Therefore, the objective of this study was to analyse the mediation of academic engagement and the dark side motivational variables between a disempowering climate and the intention to choose teaching as a career in pre-service teachers. The following hypotheses were established: (i) a disempowering climate negatively predicts the intention to become a teacher (H1); (ii) the dark side of motivation (i.e., CM, amotivation, and FBPN) negatively mediates the relationship between a disempowering climate and the intention to become a teacher (H2); (iii) academic engagement negatively preceded by the dark side of motivation (i.e., CM, amotivation, and FBPN) mediates the relationship between a disempowering climate and the intention to become a teacher (H3). The Strengthening the Reporting of Observational Studies in Epidemiology (STROBE) Initiative [39] was used for the study description.

## 2. Materials and Methods

### 2.1. Design

The study took a cross-sectional, non-randomized and descriptive design. The participating sample included students from eight Andalusian universities and the data collection process took place in May 2021. The potential participants had to meet the following inclusion criteria: (i) to be enrolled in the master’s degree in secondary and upper-secondary education teaching, vocational training and language teaching (presential modality); (ii) to be a student from one of the eight Andalusian public universities; (iii) to hand the informed consent for participation; and (iv) to complete fully the online survey questionnaire. 

### 2.2. Instruments

#### 2.2.1. Disempowering Climate

Based on the theoretical assumptions of the SDT and AGT, and previous works [19], including research on Spanish university students [21], the ego-involving climate subscales from the motivational climate in education scale [40], and the controlling style from the interpersonal teaching style in higher education scale [41] were used to measure the different dimensions. The perceived disempowering climate measure included the following two subscales: the ego-involving climate (three items, e.g., “The teacher gives most attention to the successful students”) and the controlling style (six items, e.g., “My teacher threatened to punish students to keep them in line during class”). Responses to each item were scored on a Likert scale ranging from 1 (strongly disagree) to 5 (strongly agree). The disempowering climate was calculated as the mean value of the average scores of the two factors comprising it. In the present study, the hierarchical two-factor CFA (H-CFA) model of the scale presented the following goodness-of-fit indices: χ^2^/df = 3.59, *p* < 0.001; CFI = 0.98; TLI = 0.98; RMSEA = 0.053 (90%CI = 0.042,.065), SRMR = 0.024. The reliability obtained was McDonald’s Omega (ω) = 0.88.

#### 2.2.2. BPN Frustration in Education

The Spanish version by Cuevas et al. [42] of the psychological need thwarting scale was used. In addition, the novelty frustration measure by González-Cutre et al. [43] was included. This instrument is composed of 17 items that measure autonomy (four items, e.g., “I feel pushed to behave in certain ways”), competence (four items, e.g., “Situations occur in which I am made to feel incapable”), relatedness (four items, e.g., “I feel I am rejected by those around me”), and novelty (five items, e.g., “I feel monotony”) need frustration. Responses to each item are scored on a Likert scale from 1 (strongly disagree) to 7 (strongly agree). The FBPN was calculated as the mean value of the average scores for each of the factors comprising it. In the present study, the H-CFA model of the scale presented the following goodness-of-fit indices: χ^2^/df = 4.52, *p* < 0.001; CFI = 0.95; TLI = 0.95; RMSEA = 0.062 (90%CI = 0.057,0.119), SRMR = 0.044. The reliability obtained was: ω = 0.94.

#### 2.2.3. Controlled Motivation and Amotivation

The introjected regulation, external regulation, and amotivation subscales from the Spanish version [44] of the Academic Motivation Scale [45] were used. The scale is grouped into four items per dimension to measure introjected regulation (e.g., “Because passing at university makes me feel important”), external regulation (e.g., “Because I want to have a good life in the future”) and amotivation (e.g., “I honestly don’t know; truthfully, I feel like I’m wasting my time in the master’s/undergraduate degree.”). A Likert scale ranging from 1 (Does not correspond at all) to 5 (Corresponds exactly) was used for the responses. In accordance with the SDT [26], the CM was calculated as the mean value of the average scores for external regulation and introjected regulation. In the present study, the H-CFA model of the scale presented the following goodness-of-fit indices: χ^2^/df = 4.51, *p* < 0.001; CFI = 0.97; TLI = 0.96; RMSEA = 0.062 (90%CI = 0.048,0.076), SRMR = 0.032. The reliability obtained was: ω = 0.75.

#### 2.2.4. Academic Engagement

The Spanish version for students of the Utrecht Work Engagement Student Scale (UWES-SS) was used [46]. The scale is composed of 17 items that form three dimensions: vigour (six items; e.g., “I feel strong and vigorous when I am studying or going to classes”), dedication (five items; e.g., “I am proud to follow this career”), absorption (six items; e.g., “I am immersed in my studies”). Responses to each item are scored on a Likert scale from 1 (strongly disagree) to 5 (strongly agree). Academic engagement was calculated as the mean value of the average scores for each of the factors comprising it. In the present study, the H-CFA model of the scale presented the following goodness-of-fit indices: χ^2^/df = 4.97, *p* < 0.001; CFI = 0.96; TLI = 0.94; RMSEA = 0.071 (90%CI = 0.056,0.092), SRMR = 0.052. The reliability obtained was: ω = 0.93.

#### 2.2.5. Intention to Choose Teaching as a Career

The Spanish version [8] of The Future Teaching Intention Scale (FTIS), based on Fishbein and Ajzen [47], was used. This unidimensional instrument assesses the pre-service teachers’ future intention to work as teachers and is composed of three items: “I intend to work as a teacher in the next three years”, “I will try to work as a teacher in the next three years”, and “I am determined to work as a teacher in the next three years”. Responses to each item are scored on a Likert scale from 1 (totally improbable) to 7 (extremely probable). In the present study, the CFA model presented the following goodness-of-fit indices: χ^2^/df = 1.69, *p* < 0.001; CFI = 0.97; TLI = 0.96; RMSEA = 0.035 (90%CI = 0.023,0.054), SRMR = 0.023. The reliability obtained was: ω = 0.93.

### 2.3. Procedure

The research team asked the different heads of the Schools of Education and of the master’s degree in secondary and upper-secondary education, vocational training and language teaching for their authorization and collaboration for this research. After obtaining all permissions, the potential survey respondents were contacted by email. The data collection process was conducted in May 2021 through an online survey questionnaire. The potential participants were informed on the relevance of the present study, the anonymous nature on responses, the manner to fill in the survey, as well as that the responses provided would not affect their grading in any way, and they could abandon their participation in the research at any time. The current study was approved by the Bioethics Committee of the University of Almeria (Ref: UALBIO2021/009) and followed all standards for human research from the Declaration of Helsinki. 

### 2.4. Risk of Bias

As the sampling method was for convenience, a blinding process was conducted between the participants and the researcher team responsible for data collection and analysis. To control for selection bias, the potential survey respondents participated voluntarily and anonymously, and were contacted via email.

### 2.5. Sample Size

Free Statistics Calculator v.4.0 software [48] was used to estimate the minimum sample size required to ensure the statistical power and trustworthiness of results. Under conditions of f^2^ = 0.152, statistical power of 0.99, and significant level of α = 0.05 in a structural equation model with six latent variables and 18 observable variables, a minimum sample of 1401 participants was computed. 

### 2.6. Data Analysis

Both the descriptive statistics and the correlations between the analysed variables were estimated with the SPSS program (v.28). In addition, the McDonald’s omega coefficient was calculated for each of the variables, indicating that values above 0.70 would be indicative of good reliability [48]. The SEM was controlled for sex and a two-step structural equation model was performed [49,50] with AMOS (v.26) software to analyse the predictive relationships between a disempowering climate and teaching intention through need frustration, CM, amotivation, and academic engagement. In the first step (the measurement model), the robustness of the bidirectional relationships between the variables that make up the model were analysed. In the second step, the predictive effects between the variables were examined. In the event that the multivariate normality assumption might be violated (Mardia’s coefficient = 55.65; *p* < 0.001), the analysis was performed using the maximum likelihood method and the 5000-iteration *bootstrapping* procedure [50]. The goodness of fit was evaluated with <5.0 for the chi-square coefficient and degrees of freedom (χ^2^/df), values > 0.90 for the CFI (Comparative Fit Index) and TLI (Tucker–Lewis Index), together with values as high as 0.80 for the SRMR (Standardized Root Mean Square Residual) and RMSEA (Root Mean Square Error of Approximation) [51]. To better interpret the results, the total explained variance (R^2^) was considered as a measure of the effect size [52]. Small, medium, and large effect sizes were considered to have values less than 0.02, close to 0.13, and greater than 0.26, respectively [53].

## 3. Results

### 3.1. Participants

A total of 1410 pre-service teachers participated (59.6% women; 40.3% men; 0.1% other), all of whom were enrolled in the master’s degree in secondary and upper-secondary education, vocational training and language teaching of various Andalusian public universities (Spain) (University of Almeria, 13.5%; University of Cádiz, 3.6% University of Cordoba, 7.6%; University of Huelva, 3.4%; University Jaen, 11.4%; University of Granada, 27.2%; University of Malaga, 27.6%; University of Sevilla, 5.8%), in various specialties (Physical Education, etc.). There were 26 pre-service teachers who decided not to take part in this research. The age of the participants ranged from 21 to 60 years (*M* = 23.85; *SD* = 5.13). The representative sample of pre-service teachers was 38.60% of the total population under study in accordance with official data from the eight Andalusian public universities (N = 3653) with a confidence interval of 95% and a 2.1% error rate. There were no missing values in the included sample data.

### 3.2. Preliminary Results

The descriptive statistics and the correlations between the latent study variables are presented in Table 1.

### 3.3. Main Results

In step 1, the model presented acceptable goodness-of-fit indices: χ^2^/df = 5.193, *p* < 0.001; CFI = 0.94; TLI = 0.93; RMSEA = 0.068 (90%CI = 0.063; 0.073), SRMR = 0.062. In step 2, the predictive SEM model showed similar goodness-of-fit indices: χ^2^/df = 5.193, *p* < 0.001; CFI = 0.94; TLI = 0.93; RMSEA = 0.068 (90%CI = 0.063; 0.073), SRMR = 0.062. Although the χ^2^/df ratio is greater than 5.0, we must consider that the chi-square test of exact fit often rejects the null hypothesis, especially in large samples, even when the postulated model is only trivially false [54]. The explained variance achieved was 33% for amotivation, 32% for FBPN, 16% for CM, 26% for academic engagement, and 21% for intention to choose teaching. After controlling for sex, in the SEM, the direct relationships between a disempowering climate and the three dark-side variables were positive and significant (strong with amotivation and FBPN; moderate with CM). Likewise, although the direct relationship between a disempowering climate and engagement was not significant, the relationship between a disempowering climate and the intention to choose teaching was significant and positive (low effect). In contrast, the other three dark-side variables showed significant direct effects on engagement: CM had a moderate positive effect, while FBPN and amotivation had negative effects. Of these three variables (i.e., CM, FBPN, and amotivation), only amotivation was significantly and negatively related (a moderate effect) to the intention to choose teaching. Finally, the direct effect of engagement on the intention to choose teaching was significant and positive, although low (see Figure 2).

Regarding indirect effects, only amotivation mediated between a disempowering climate and the intention to choose teaching, although with a significant negative relationship (−0.23). The overall indirect effects between a disempowering climate and engagement via the dark-side variables were not statistically significant (*p* = 0.424). The total effects of a disempowering climate on the intention to choose teaching were only increased through CM and academic engagement (0.24), whereas they decreased through FBPN and engagement (0.18), and amotivation and engagement (0.19), and were even negative when only considering the effect of amotivation (−0.02). Lastly, there was a statistically significant overall indirect effect between a disempowering climate and the intention to choose teaching (β = −0.29; CI95% = −0.22; −0.36; *p* = 0.12).

## 4. Discussion

The objective of this research was to analyse the mediating role played by academic engagement and the dark-side motivational variables between a disempowering motivational climate and the intention to choose teaching as a career in the context of initial teacher training. The main results highlight the importance of the mediation from the dark side of motivation, through academic engagement, between a disempowering motivational climate and the intention to choose teaching as a career. 

In accordance with the hypothesized model, the results reveal that a disempowering motivational climate used by teacher trainers had a direct positive effect on the teaching intention of pre-service teachers (rejecting H1). These findings are not in line with the postulates of the theory of planned behaviour [37]; that is to say, following its theoretical foundation, a disempowering climate would influence the perceived behavioural control of pre-service teachers, triggering a decrease in the antecedent of future behaviour (i.e., the intention to teach). However, the present findings depart from this theoretical foundation. In other words, pre-service teachers who perceive that their educators create a disempowering motivational climate will not have their intention to choose teaching as a career diminished, but quite the opposite. This contradictory result might be due to the effect that this multidimensional conceptualization has [17] on the motivational variables. Following the fundamentals of SDT (social factors → psychological mediators → motivation → consequence; see [15]), socio-psychological factors do not present a direct relationship on educational outcomes; if not, they are influenced by the motivational consequences that the social factors generate. 

Continuing with the hypothesized model, the results reveal that the perception of a disempowering motivational climate has a direct positive effect on the dark side of motivation and, through amotivation, has a negative effect on the intention to teach (H2). In addition, only amotivation (from among the dark-side motivational outcomes used) was related to the intention to choose teaching as a career in pre-service teachers. These results contrast with other studies, such as those by Trigueros et al. [24] and Viksi and Tilga [25], and underline that neither CM nor FBPN predict the behavioural intention of students without the mediation of cognitive elements (e.g., academic engagement). A disempowering context in which students feel that their BPNs are frustrated, or feel controlled by external pressures, will not diminish their intention to choose teaching as a career. In contrast, a disempowering socio-contextual climate that promotes a lack of motivation will reduce the intention of pre-service teachers to become teachers. This might be because a lack of motivation, regardless of its causes (e.g., believing one has poor ability or poor effort, insufficient academic values, or unattractive schoolwork characteristics; see Green-Demers et al. [55]), has an influence on later teaching behaviours [23]. Specifically, amotivation is related to the lowest level of self-determination and represents a total absence of will with respect to antecedent behaviour (i.e., the intention to teach) [56]; that is to say, when pre-service teachers perceive a motivational classroom climate characterized by high degrees of disempowerment, their intention to become teachers will decrease.

Finally, the SEM showed that academic engagement preceded by the dark side of motivation acted as a mediator between a disempowering motivational climate and the intention to choose teaching as a career (H3). Although all three relationships were statistically significant, it is worth noting the negative and significant effect of amotivation and FPBN on academic engagement and the intention to teach. These results are consistent with the studies by Trigueros et al. [24] that examined the dark side through cognitive elements on educational outcomes and, in turn, with the results of Viksi and Tilga [25], which stated that the dark side influences future intention behaviours. Pre-service teachers who perceive a disempowering climate, capable of frustrating their BPNs or reducing their self-determined motivation, reduce their academic engagement and, consequently, their intention to choose teaching as a career. This can be explained by the influence of cognitive variables, such as academic engagement, on motivational processes and future intention behaviours [57]. Furthermore, in the SEM, a positive and significant predictive relationship of CM was found on academic engagement and the intention to teach based on a disempowering climate: if the disempowering climate generated by the teacher produces a behaviour based on external contingencies (i.e., CM), the pre-service teacher will increase their academic engagement and, as a consequence, increase their intention to choose teaching as a career. These results are far from the SDT [15] and other previous studies [23]. However, as Howard et al. [58] pointed out in a recent meta-analysis, a possible explanation for the above observation may be the influence of introjected regulation within the motivational processes of self-determination in the educational context. Likewise, introjection represents a partial internalization of self-determined values and, as such, can drive behaviours through ego-involvement and the use of internalized pressures [15], in such a way that, in the pre-service teachers, the internalization of controlled forms of motivation induced by a disempowering motivational climate would provoke a regulation of behaviours alien to the student, capable of favouring the development of adaptive cognitive processes. Although authors such as Howard et al. [58] have evidenced the double role (adaptive and maladaptive) of the influences of external motivations, far from the postulates of the SDT, there is no specific scientific evidence in the educational field that provides a specific explanation for this incongruity. Therefore, future studies should aim to establish a relationship between the dark side of motivation and a multidimensional perspective of academic motivation [59]. 

### 4.1. Limitations and Future Perspectives

Based on the results presented, this research clearly contributes to developing Duda and Appleton’s [17] conceptualization of the dark motivational pathway, introducing academic engagement and demonstrating its influence on the intention to choose teaching as a career. The model helps to understand how characteristic elements of a disempowering climate might influence FBPN to the extent that aspects of a need-thwarting environment could be considered. Another strength of this study is that direct and mediated relationships are established that deepen and expand our knowledge regarding maladaptive behaviours on the dark side of motivation. Despite the above findings, the present research also has certain limitations. First, the convenience sampling method used means the results obtained should be interpreted with caution. Second, the cross-sectional design of the research only represents a particular view in time, neglecting the possibility that causal relationships might be established between a disempowering motivational climate and the motivational outcomes. Therefore, longitudinal research is needed to examine the fluctuations that the social-psychological environment and the dark side of motivation have on the cognitive consequences of students throughout the teacher training process. Third, the research based its main conclusions on self-reported questionnaires completed by the pre-service teachers. Future research should examine the disempowering motivational climate using different instruments and taking into account the teachers’ perceptions so as to triangulate the data. Lastly, although the present research focused on examining the consequences of a disempowering climate and the dark side of motivation, it only evaluated positive educational consequences (i.e., the intention to become a teacher and academic engagement). Therefore, future research should evaluate the influence of the dark side of motivation over a broader spectrum of maladaptive consequences (e.g., disengagement and burn out) [60]. In this way, the consequences of the dark side of motivation on educational outcomes, and their role in the educational field, could be understood in more detail and depth. 

### 4.2. Practical Implications for Initial Teacher Training

From the results of the present research, certain educational implications of interest can be established, especially for teacher educators trying to encourage their students to participate in learning [61]. While previous research has shown the importance of generating an empowering motivational climate [19], the present results demonstrate the effects of a disempowering motivational climate on educational outcomes. Therefore, teacher educators should avoid strategies that create a disempowering climate [17,23], such as:-Establishing threats regarding deadlines or strict rules (e.g., “Those not attending 85% of the subject will not be eligible for evaluation”).-Using normative behaviours in the classroom (e.g., “You cannot leave the classroom until the explanation has been given”).-Employing destructive criticism (i.e., “You have to know the characteristics of your students if you want to teach correctly in future”).-Denying the students’ pedagogical contributions (e.g., “No, the report should be this length and have these characteristics”).-Using explicitly controlling language, such as “you must” or “you have to” (e.g., “You have to submit both tasks in order to take the exam”).

In addition, a series of considerations are proposed to help teacher educators avoid generating a disempowering climate [16,17,23]: (i) The generation of a disempowering climate does not mean that teacher educators should refrain from structuring the learning process. They can start by establishing expectations or providing a scaffold while supervising the learning process of pre-service teachers [62,63]; that is, establishing the use of norms through understanding (i.e., why is a standard introduced?) and establishment (i.e., participating in the standard-setting process with the students); (ii) The perceptions of the teacher educators regarding the classroom climate differ from those of the students [16,64,65]. Teacher educators may have an erroneous perception of the classroom climate they are trying to avoid, which conflicts with the students’ perception [66]. For this, various tools such as class video recording or student self-reports [67] can be used to obtain an overview of the socio-environmental climates generated; (iii) As has been shown, some strategies for controlling the pre-service teachers’ motivation can produce behavioural benefits, such as developing academic engagement and the intention to become a teacher. These outcomes are the result of specific motivational experiences caused by disempowering climates. In fact, the associated behavioural consequences of a dominating climate can exact an emotional cost, reducing psychological well-being over the long term [23].

## 5. Conclusions

The findings from the present study reveal the association between a disempowering motivational style and its influence on the dark side of motivation, which act as negative promoters on the intention of pre-service teachers to pursue a teaching career. In addition, the results reveal the negative mediating influence that academic engagement has as a consequence of the dark side of motivation on the intention to be a teacher. Finally, the study findings highlight the importance of CM on the academic engagement of trainee teachers as a positive mediator between a disempowering motivational climate and the intention to become a teacher. In this regard, teacher educators should avoid a disempowering motivational climate given its negative consequences on academic engagement through the dark side of motivation, except when students perceive external demands during the teacher training process.

## Figures and Tables

**Figure 1 ijerph-20-00878-f001:**
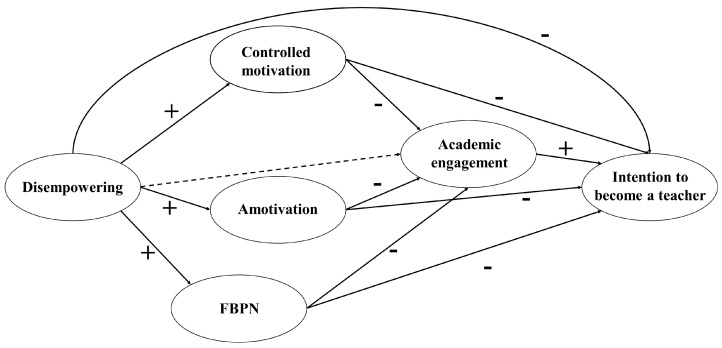
Hypnotized model. Note: the dashed lines represent non-significant relationships; FBPN = frustration of basic psychological needs.

**Figure 2 ijerph-20-00878-f002:**
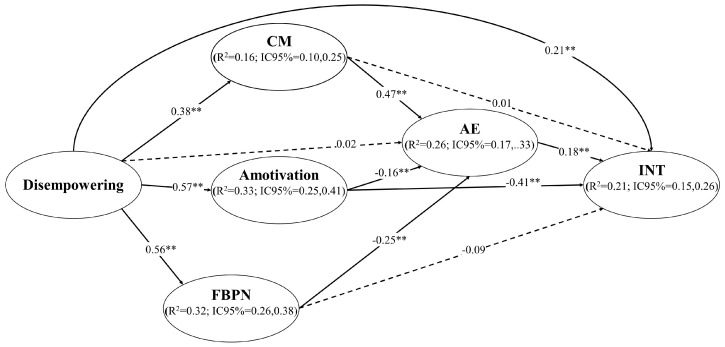
Predictive relationships between a disempowering motivational climate and the intention to choose teaching through the mediating role of controlled motivation, frustration of basic psychological needs, amotivation, and academic engagement. Note: ** *p* < 0.001. CM = Controlled motivation; FBPN = Frustration of Basic Psychological Needs; AE = Academic engagement; INT = intention to choose teaching; R^2^ = Explained variance; CI = Confidence interval. The dashed lines represent non-significant relationships.

**Table 1 ijerph-20-00878-t001:** Descriptive statistics and correlation between variables.

Variable	*M*	*SD*	Q1	Q2	ω	2	3	4	5	6
1.Disempowering	2.34	0.87	0.51	−0.18	0.88	0.31 **	0.45 **	0.43 **	−0.05	−0.07 *
2.Controlled motivation	3.22	0.77	0.02	−0.22	0.75		0.22 **	0.17 **	0.24 **	0.09 **
3.FBPN	2.41	0.85	0.33	−0.37	0.94			0.47 **	−0.23 **	−0.21 **
4.Amotivation	2.07	0.92	0.74	−0.02	0.73				−0.30 **	−0.24 **
5.Academic Engagement	3.47	0.82	−0.27	−0.09	0.93					0.26 **
6. Intention to choose teaching	5.99	1.38	−1.41	1.52	0.93					

Note. ** The correlation is significant at the 0.01 level; * The correlation is significant at the 0.05 level. *M* = mean; *SD* = standard deviation; Q1 = skewness; Q2 = Kurtosis; ω = McDonald’s omega; FBPN = Frustration of the Basic Psychological Needs.

## Data Availability

The data presented in this study are available on request from the corresponding author. The data are not publicly available due to privacy.

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
