# Peer review of "Detrimental Effects of Disempowering Climates on Teaching Intention in (Physical Education) Initial Teacher Education"

_ijerph, 2023, doi:10.3390/ijerph20010878_

Round 1

Reviewer 1 Report

The manuscript “Detrimental effects of teacher educator-created disempowering climates on the teaching intention of pre-service teachers? Focus on the dark motivational path” explores the dark side of Duda's multidimensional conceptualization, its influence on academic engagement, and the intention of pre-service teachers to be educators. The topic is not directly related to the priorities of " International Journal of Environmental Research and Public Health" journal, however, it is an acceptable topic.
In general, the manuscript is well organized and presents a clear structure. The ethics statements and data availability statements are adequate. The manuscript is scientifically sound, and the experimental design is appropriate to test the hypothesis. I agree with the methodological approach followed. The results are consistent with the nature of research objectives. The conclusions are consistent with the evidence and arguments presented.
The overall paper evaluation is positive and reveals potential for publication with minor revisions:
- the title is too long, should be shorter;
- [351-355] lines must be removed.

Author Response

Manuscript ID: ijerph-2064248

We thank the reviewer for his/her constructive comments and his/her thorough revision of the manuscript. Below we answer his/her questions and concerns, including explicitly the changes made in the manuscript as well.

Reviewer - 1

Comments and Suggestions for Authors

The manuscript “Detrimental effects of teacher educator-created disempowering climates on the teaching intention of pre-service teachers? Focus on the dark motivational path” explores the dark side of Duda's multidimensional conceptualization, its influence on academic engagement, and the intention of pre-service teachers to be educators. The topic is not directly related to the priorities of " International Journal of Environmental Research and Public Health" journal, however, it is an acceptable topic.

In general, the manuscript is well organized and presents a clear structure. The ethics statements and data availability statements are adequate. The manuscript is scientifically sound, and the experimental design is appropriate to test the hypothesis. I agree with the methodological approach followed. The results are consistent with the nature of research objectives. The conclusions are consistent with the evidence and arguments presented.

Comment: The title is too long, should be shorter

  • Response: Following the reviewer's suggestions, the title has been revised and shortened.

Comment: [351-355] lines must be removed

  • Response: Suggestions made by the reviewer were made.

Reviewer 2 Report

This is a potentially interesting study on the impact of a dis-empowering motivational style and its influence as a negative promoter.

However, in the present version, the manuscript has a weak point, which needs to be corrected:

A description of population (is it students in Andalusian region? then, where Andalusian region is and its characteristic) related to the sample should be included in the method section, in order to assess the representativeness of the chosen sample. Without this information, the study could not be interpreted by readers outside Spain.

Author Response

Manuscript ID: ijerph-2064248

We thank the reviewer for his/her constructive comments and his/her thorough revision of the manuscript. Below we answer his/her questions and concerns, including explicitly the changes made in the manuscript as well.

Reviewer – 2

Comments and Suggestions for Authors

This is a potentially interesting study on the impact of a dis-empowering motivational style and its influence as a negative promoter.

However, in the present version, the manuscript has a weak point, which needs to be corrected:

Comment: A description of population (is it students in Andalusian region? then, where Andalusian region is and its characteristic) related to the sample should be included in the method section, in order to assess the representativeness of the chosen sample. Without this information, the study could not be interpreted by readers outside Spain

 Response: Upon the reviewer’s request, we have detailed that the participants comprised a representative sample by depicting 38.60% of the population under study according to official data from the Andalusian public universities (N=3,653). The sample size was estimated with a confidence interval of 95% and a 2.1% bias rate (see, paragraph “Participants”, lines 283-295).

Reviewer 3 Report

This is a well-established article. I am impressed by the way of presenting the theoretical framework and research methods. The only minor concern here is the way of dealing with missing data. If missing data were simply deleted, it is better to provide an explanation of it. Otherwise, it is suggested to use imputation to identify potential patterns in the missing data.

Author Response

Manuscript ID: ijerph-2064248

We thank the reviewer for his/her constructive comments and his/her thorough revision of the manuscript. Below we answer his/her questions and concerns, including explicitly the changes made in the manuscript as well.

Reviewer – 3

Comments and Suggestions for Authors

This is a well-established article. I am impressed by the way of presenting the theoretical framework and research methods. The only minor concern here is the way of dealing with missing data. If missing data were simply deleted, it is better to provide an explanation of it. Otherwise, it is suggested to use imputation to identify potential patterns in the missing data.

Comment: It is better to provide an explanation of completely deleting cases with missing data. Otherwise, it is better to use missing data imputation to deal with any potential patterns existing in the cases with missing data.

  • Response: According to the comment made by the reviewer, we would like to emphasize that the online questionnaire was administrated via Google Form®. Under the conditions of this type of format, the survey respondents were obligated to respond all items, if they wanted to fully complete the questionnaire. For this reason, there were no missing values (see line 295). Moreover, we should point that a total of 26 pre-service teachers decided not to take part in this research by not delivering the informed consent (see lines 290-291).